# Major depressive disorder and irritable bowel syndrome risk: A Mendelian randomization study

**Guowei Sun[1], Yueyi Jiang[2]***

**1** Department of General Surgery, Traditional Chinese Medicine Hospital of Yangzhou, Yangzhou, China,
**2** Department of Anesthesiology, The Affiliated Cancer Hospital of Nanjing Medical University, Nanjing, China

* jyy2021121289@stu.njmu.edu.cn

## Abstract

### Background

Previous studies have revealed a connection between major depressive disorder (MDD) and irritable bowel syndrome (IBS), but it remains obscure if the two diseases are related causally. Mendelian randomization was utilized in this investigation to ascertain whether MDD contributed to the emergence of IBS.

### Methods

To examine possible connections between MDD and IBS, we used two-sample Mendelian randomization (MR) utilizing summary data from genome-wide association studies (GWAS). The Psychiatric Genomics Consortium (PGC) provided information on genetic associations with MDD (cases: 135,458; controls: 344,901). The Medical Research Council Integrative Epidemiology Unit (MRC-IEU) provided information on genetic associations with IBS (cases:10,939; controls:451,994). Inverse Variance Weighted (main analyses), MR-Egger regression, Weighted mode, and Weighted Median were the four MR methods used in this investigation. In addition, we also performed multiplicity and heterogeneity analyses to eliminate possible biases.

### Results

In the standard Inverse Variance Weighting (IVW) method, an increased risk of IBS was linked to a genetic susceptibility to MDD (OR: 1.01; 95% CI: 1.006 to 1.014, $p = 1.02E-07$). In addition, neither significant heterogeneity (IVW Q = 24.80, $p = 0.73$) nor horizontal pleiotropy (MR Egger $p = 0.17$; MRPRESSO $p = 0.54$) were detected in this MR analysis. The bidirectional analysis, however, did not show a genetic link between IBD and MDD ($p$ steiger <0.01).

### Conclusion

A direct causal relationship between MDD and IBS was revealed by Mendelian randomization study, which contributes to the effective clinical management of both diseases.

**Data Availability Statement:** Data including original data, figures, tables, and Supporting information files are available at: https://doi.org/10.6084/m9.figshare.25131929.v3.

**Funding:** The author(s) received no specific funding for this work.

**Competing interests:** The authors have declared that no competing interests exist.

**Abbreviations:** GWAS, Genome-Wide Association Studies; IBS, Irritable bowel syndrome; IVs, Instrumental variables; IVW, Inverse Variance Weighting; LD, Linkage disequilibrium; MAF, Minor Allele Frequency; MDD, Major Depressive Disorder; ML, Maximum Likelihood; MR, Mendelian randomization; SNPs, Single nucleotide polymorphisms; TSMR, Two-sample MR; WM, Weighted Median.

## 1. Introduction

Major depressive disorder (MDD), which has a lifetime prevalence of 15–30%, is a very common mental health condition that is predominantly characterized by depressed mood, decreased interest, ruminative thoughts, loss of pleasure, feelings of guilt or worthlessness, decreased energy, poor cognition, vegetative symptoms, and suicidal attempts. About twice as many women as males have MDD, and MDD affects above 6% of the global adult population every year [1]. Irritable bowel syndrome (IBS) symptoms are prevalent in those who have been given a diagnosis of depressive disorders, according to reports [2, 3]. Irritable bowel syndrome (IBS), one of the most prevalent functional gastrointestinal disorders, is characterized by abdominal pain, bloating, and changes in bowel patterns (constipation, diarrhea, or both), but there are no known structural abnormalities [4]. The total prevalence of IBS in the world is 11.2% [5–7].

Both MDD and IBS have been reported to cause significant physical and psychological harm to a significant population. Many patients have a significant decline in quality of life, and society bears a hefty financial burden [8, 9]. Irritable bowel syndrome, a major public health problem, can have a significant financial burden. According to a systematic study released in 2013, the cost of irritable bowel syndrome treatment ranges from $1,562 to $7,547 per year for direct costs and from $791 to $7,737 annually for indirect costs in the United States [10]. Not only that, but IBS also results in humanistic burdens including reduced personal quality of life, decreased socialization and insufficient rest [11].

A link between IBS and MDD has been suggested by earlier investigations. A cross-sectional study that was conducted more recently on the incidence of IBS symptoms in patients with MDD discovered that these patients had higher rates of IBS symptoms than those in the control group [12]. However, the relationships between genetic and environmental exposures and other potential confounding factors in human observational research also present a number of additional difficulties. Thus, understanding the connection between the two concurrent disorders may be of utmost importance to public health as a result of the disputed causal association between MDD and IBS.

MR is a useful method for assessing causal relationships, which has the advantage of using genetic variation known as single nucleotide polymorphism (SNP), which is randomly assigned at conception, thus exactly simulates the randomization process of randomized controlled trials, so as to eliminating environmental exposure, socio-economic conditions, behavior and reverse causality and other confounding variables [13, 14].

In order to determine whether there is a causal connection between the two diseases, we first hypothesized that IBS might be a downstream effect of MDD. We then chose SNPs from GWAS and used two-sample MR (TSMR), a technique that does not require individual-level data, to infer the relationship.

## 2. Methods

### Mendelian randomization and assumptions

Our study utilized the TSMR method to examine the relationship between MDD and IBS. We based our MR analysis on three fundamental hypotheses (Fig 1): (1) Correlation hypothesis: The chosen genetic variant should demonstrate a strong association with the exposure variable. Weak instruments may result in biased estimates; (2) Independence hypothesis: This hypothesis suggests that the genetic variant used as an instrument for the exposure variable is associated with the outcome solely through its impact on the exposure, without any alternative pathway involvement. In essence, the genetic variant should not be linked to any confounders

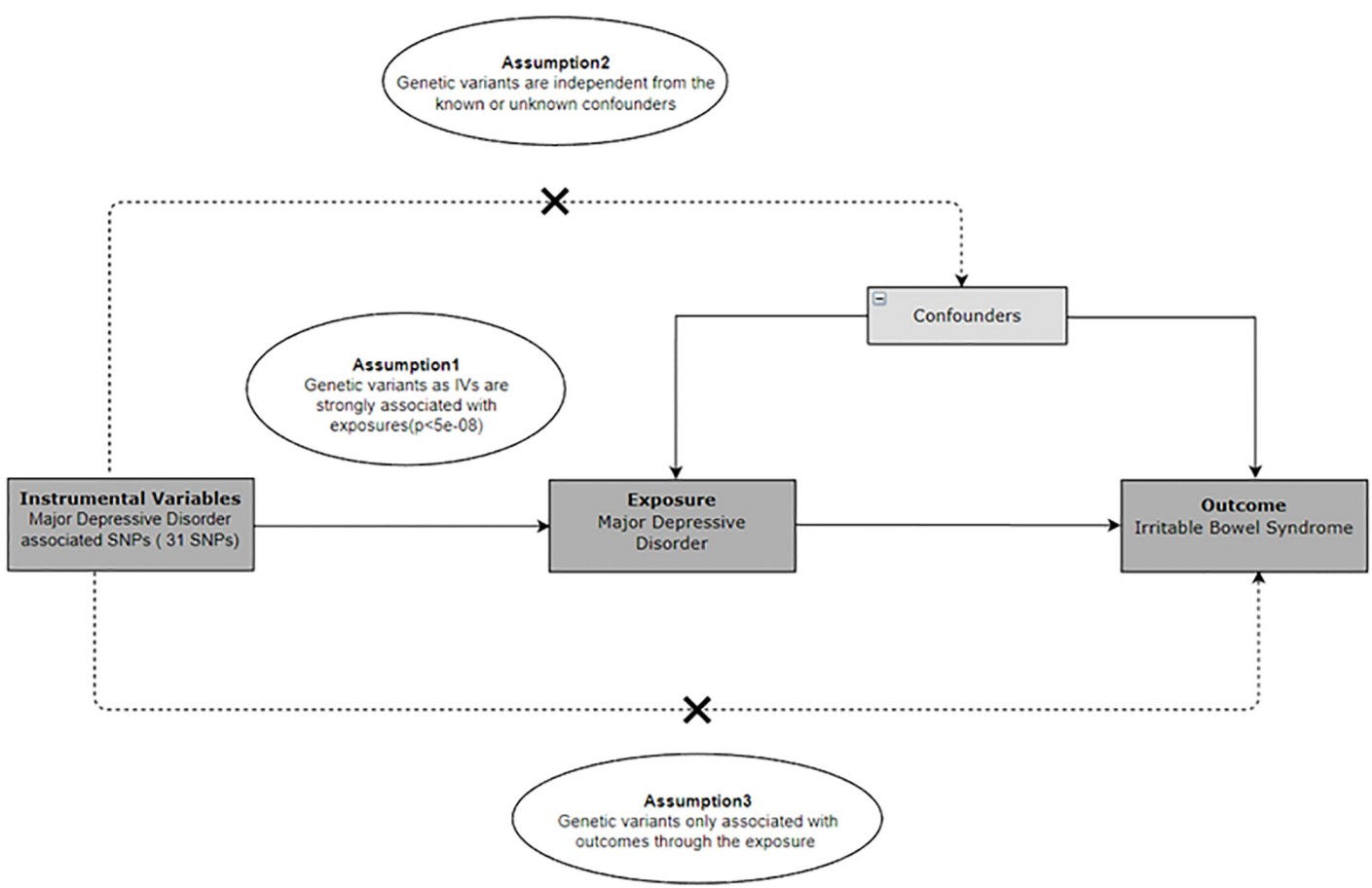

**Fig 1. Diagram of the Mendelian randomization assumptions supporting a two-sample Mendelian randomization analysis of the causal effect of major depressive disorder on irritable bowel syndrome.**

of the exposure-outcome relationship; (3) Exclusion hypothesis: This hypothesis asserts that the genetic variant used as an instrument influences the outcome exclusively through its effect on the exposure, without involvement in any other pathway. Violation of this assumption, such as pleiotropy (wherein the genetic variant affects multiple phenotypes), can lead to biased estimates. In traditional observational studies, confounders are variables associated with both the exposure and outcome of interest. Failure to properly control for confounders can result in biased estimates of the exposure-outcome relationship. Mendelian randomization mitigates confounding factors by employing genetic variants as instrumental variables (IVs). These IVs are randomly allocated during meiosis and are less susceptible to environmental or lifestyle influences. This approach mimics a randomized controlled trial, leveraging the natural randomization of genetic inheritance to reduce bias from confounding variables.

## Data resources

The Psychiatric Genomics Consortium (PGC) is recognized as a pioneering initiative in psychiatry, fostering collaboration and accelerating progress in unraveling the genetic basis of psychiatric disorders. With over 800 investigators spanning 36 countries and a participant cohort exceeding 400,000 individuals, the PGC represents a global effort to advance mental health research. The SNPs related to MDD were identified from the 135,458 patients and

344,901 population controls in the Psychiatric Genomics Consortium (PGC) (PGC—Psychiatric Genomics Consortium (unc.edu)). The Medical Research Council (MRC) Integrative Epidemiology Unit at the University of Bristol (MRC IEU) aims to integrate molecular, cellular, clinical, and population data to identify causal associations between modifiable exposures and health outcomes. Employing a multidisciplinary approach, the Unit addresses key public health issues pertinent to both the UK and global agendas. By leveraging disciplines beyond traditional epidemiology, the MRC IEU enhances causal and translational epidemiology. The SNPs linked to IBS were found in the Medical Research Council Integrative Epidemiology Unit (MRC-IEU) (MRC Integrative Epidemiology Unit—UKRI) dataset of 10,939 patients and 451,994 controls (S1 Table). In the analysis to study the causal impact of MDD on IBD, we determined 31 single nucleotide polymorphisms (SNPs) as potential genetic instruments (S2 Table).

## SNP selection

To ensure the authenticity and accuracy of the conclusions regarding the causal relationship between MDD and IBS, rigorous quality control measures were implemented to select optimal instrumental variables (IVs). Firstly, SNPs significantly associated with MDD were chosen as IVs. Two thresholds were applied for IV selection. Initially, SNPs with p-values below the genome-wide statistical significance threshold (5.00E-08) were selected to serve as IVs. Secondly, a minor allele frequency (MAF) threshold of 0.3 was applied to the variants of interest. Thirdly, adherence to one of the principles of the MR approach, which dictates that there should be no linkage disequilibrium (LD) among the included IVs, was ensured. To assess LD between the included SNPs, a clumping process was conducted ($R^2 < 0.001$ and clumping distance = 10,000 kb). Fourthly, a crucial step in MR analysis is to confirm that the effects of the SNPs on the exposure align with the same allele as the effects on the outcome. To prevent distortion of strand orientation or allele coding, palindromic SNPs (e.g., those with A/T or G/C alleles) were excluded.

## Mendelian randomization analyses

The inverse variance weighted (IVW) method estimates a causal effect by calculating the slope of a regression line over the associations between the weighted SNP-mean exposure and SNP-mean outcome (oriented to be positive) [15]. Statistically significant heterogeneity was considered when the Cochran's Q test *p* was <0.05, the random-effects IVW model was adopted regardless of heterogeneity. In addition, a sensitivity analysis was conducted to assess potential bias resulting from directional pleiotropy or heterogeneity of individual SNP IVs. This involved employing MR-Egger regression, weighted median, and weighted mode MR methods as supplements to the IVW method [16]. MR-Egger regression can accurately estimate MR estimates in the presence of horizontal pleiotropy and quantify the degree of bias produced by this phenomenon. The intercept was utilized to determine whether directional pleiotropy has an impact on the causal estimates [17]. Weighted median method can provide a consistent MR estimate of causal influence if at least half of the instrumental variables are invalid. Weighted mode method can offer a robust causal estimate when the majority of individual estimates were from valid IVs [15]. Additionally, to identify potentially influential SNPs, which could be driven for example by horizontal pleiotropy, we performed a "leave-one-out" sensitivity analysis in which we sequentially omitted one SNP at a time. We used the funnel plot, a commonly used graphic test,to evaluate possible directional pleiotropy [18]. Finally, to eliminate the horizontal pleiotropy, we used the MR Pleiotropy Residual Sum and Outlier (MR-PRESSO) test to determine and eliminate the pleiotropic effects caused by outliers [19]. Analyses were

implemented by the package TwoSampleMR (version 0.5.6) [18] and MRPRESSO (version 1.0) [19] in R (version 4.0.1). No additional ethical approval was required due to the re-analysis of previously collected and published data.

## 3. Results

In this study, instrumental variables (IVs) were selected using specific criteria. Single nucleotide polymorphisms (SNPs) associated with each trait at the locus-wide significance threshold ($p < 5.00E-08$) were considered as potential IVs to ensure a robust correlation between genetic variants and the exposure variable. LD between SNPs was evaluated using data from the 1000 Genomes Project European samples, and only SNPs with an LD value of $R^2 < 0.001$ (clumping window size = 10,000kb) and the lowest $P$-values were retained. Additionally, SNPs with a MAF $> = 0.3$ were excluded to ensure the independence of the genetic variables. Ultimately, 31 statistically significant SNPs were identified from GWAS summary data for MR analysis (S2 Table).

The main analysis was conducted using the inverse variance weighted (IVW) method, which estimates a causal effect by calculating the slope of a regression line over the weighted SNP-mean exposure vs SNP-mean outcome associations (orientated to be positive). IVW proposed a causal link between MDD and IBS (OR: 1.01; 95% CI: 1.006 to 1.014, $p = 1.02E-07$).

Mendelian randomization is primarily used to assess causal relationships rather than directly determine the magnitude of quantitative effect sizes. While Mendelian randomization can provide insights into the direction and plausibility of the causal impact of an exposure on an outcome, it often cannot precisely measure the magnitude of this effect. Mendelian randomization utilizes genetic variants to randomly allocate exposure, mimicking the design of a randomized controlled trial. Because genetic variants are randomly assigned during the process of inheritance, their relationship with the exposure is less likely to be influenced by confounding factors. Therefore, Mendelian randomization studies can provide more reliable inference about the causal impact of exposure on the outcome. Although Mendelian randomization can offer clues about causal relationships, its results typically do not directly yield quantitative effect sizes. Assessing the magnitude of effect sizes may require incorporating other types of study designs, such as cohort studies or randomized controlled trials, to obtain more precise estimates. Thus, when interpreting Mendelian randomization study results, the emphasis is usually placed on evaluating the presence and direction of causal relationships rather than directly quantifying the size of effects.

We also performed MR-Egger, Weighted median, and Weighted mode for sensitivity analysis to verify the consistency of test results (Table 1). The MR-Egger method tested the null association hypothesis and provided an effect estimate, considering potential invalid SNPs [17]. The weighted median approach, providing a consistent estimate even with invalid SNPs [15]. The weighted mode method can offer a robust causal estimate when the majority of individual estimates were from valid IVs [15]. Method comparison plot (Fig 2) represents the results of Mendelian randomization (MR) analysis between MDD and IBS. Each data point

**Table 1. MR estimates from each method of assessing the causal effects of major depressive disorder on irritable bowel syndrome.**

| Method | nSNP | b | se | p | Odds Ratio(95% CI) | Q(p) | Egger-intercept(p) |
|---|---|---|---|---|---|---|---|
| MR Egger | 31 | 0.0281 | 0.0128 | 0.0368 | 1.03(1.003–1.055) | 22.81(0.79) | |
| Inverse variance weighted | 31 | 0.0102 | 0.0019 | 1.02E-07 | 1.01(1.006–1.014) | 24.80(0.73) | -0.0006(0.1703) |
| Weighted median | 31 | 0.0114 | 0.0028 | 3.8E-05 | 1.01(1.006–1.017) | | |
| Weighted mode | 31 | 0.0165 | 0.0055 | 0.0052 | 1.02(1.006–1.028) | | |

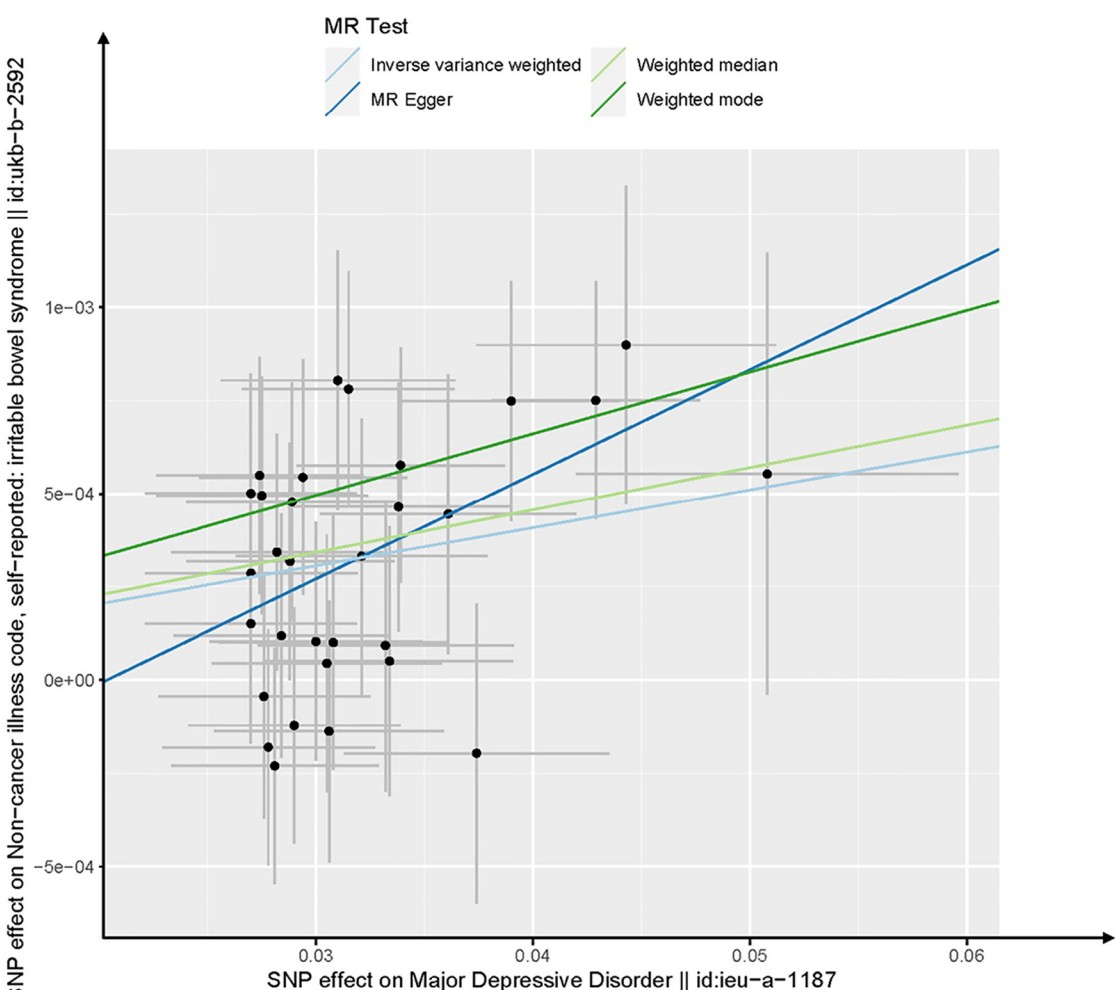

**Fig 2. Scatter plots of the genetic causal associations with major depressive disorder against irritable bowel syndrome using different MR methods.**

corresponds to a SNP serving as an instrumental variable, with the accompanying line indicating the 95% confidence interval. The x-axis denotes the SNP's effect on the exposure factor (MDD), while the y-axis indicates its effect on the outcome factor (IBS). The slope of the colored lines reflects the ratio of these effects, representing the exposure's impact on the outcome. Different colors represent distinct algorithms. Overall, the lines exhibit an upward trend, suggesting that as MDD increases, the risk of IBS also rises. In the final step, we conducted leave-one-out sensitivity analysis to determine if the association between MDD and IBS was disproportionately influenced by any single SNP. The resulting forest plot was then examined (S2 Fig). Each black point in the forest plot represents the MR analysis (using IVW) excluding that particular SNP. Furthermore, the overall analysis including all SNPs is presented for comparison. The forest plot revealed a reliable and stable outcome, indicating that the causal effect of MDD on IBS remained consistent and was not significantly affected by excluding any single SNP.

In order to conduct quality control, in addition to performing the leave-one out sensitivity, we also conducted horizontal polymorphism and heterogeneity analysis. The MR-Egger intercept test evaluated pleiotropy, with a significant difference from zero indicating potential

horizontal pleiotropy. If SNPs influence the outcome through a pathway other than the exposure, it constitutes horizontal pleiotropy, violating Mendelian randomization (MR) assumptions. Consistent bias in MR estimates can result if the average horizontal pleiotropic effect of SNPs favors one direction. The Egger regression intercept provides an estimate of horizontal pleiotropy magnitude. We discovered no measurable evidence of horizontal pleiotropy in MR-Egger regression and MR-PRESSO analysis, which were employed to test for and account for violations of the horizontal pleiotropy assumption of Mendelian randomization (MR-Egger $p = 0.17$; MR-PRESSO $p = 0.54$). When using a fixed effect model, IVW makes the assumption that there is no horizontal pleiotropy. When using a random effect model, IVW makes the assumption that (1) the strength of association of the genetic instruments with the risk factor is not correlated with the magnitude of the pleiotropic effects and (2) the average pleiotropic effect is zero. Since there was no statistical horizontal pleiotropy, we used a fixed- effect model. Heterogeneity pertains to the variability in causal estimates obtained for each SNP, indicating the consistency of the causal estimate across all SNPs. Lower heterogeneity implies greater reliability of Mendelian randomization (MR) estimates. Heterogeneity was evaluated using each applicable MR method. Heterogeneity between SNPs was assessed using Cochran's Q statistics. The Cochrane Q test for heterogeneity indicated that the studies are not heterogeneous. (IVW Q = 24.80, $p = 0.73$). The funnel plots also suggest that heterogeneity among SNPs is unlikely (S1 Fig). The aforementioned findings show that MR analysis has strong estimation and weak bias.

## 4. Discussion

Patients with MDD are at a significant risk for developing IBS, although the exact cause of this risk has not yet been determined. Our study is the first, as far as we are aware, to use MR analysis and large-scale GWAS data to show the causal link between MDD and IBS. Scholars have hypothesized that there is a brain gut axis through which intestine affects brain health. Our findings support past findings of observational studies that gut and brain processes interact pathophysiologically, revealing the existence of the gut-brain axis. The central and enteric neural systems communicate in both directions through the gut-brain axis, which connects the brain's emotional and cognitive regions with the peripheral activities of the gut. The gut-brain axis, which includes the enteric nervous system, the central nervous system, the autonomic nervous system, and the neuroendocrine and neuroimmune systems, is an intricate web of linked neural connections that connect the gut and the brain [20]. Both genetic and environmental influences on brain development are mediated by it. Inspired by the brain gut axis mechanism, we conducted an extensive MR investigation to determine the cause-and-effect relationship between MDD and IBS.

The causal link between MDD and IBS was found using the four MR methods: IVW, MR-Egger regression, Weighted mode, and Weighted Median. Given that all four techniques pointed in the same way, we draw the conclusion that MDD and IBS were related causally.

Major depressive disorder (MDD) and irritable bowel syndrome (IBS) frequently occur together. It is important to clarify the causal relationship between the two diseases and to explore the underlying mechanisms, which can aid with prevention and control. On the one hand, patients with MDD should prioritize is preventing IBS. About 20% to 30% of IBS patients have comorbid MDD [21]. Between 20% and 30% of people with IBS also have concomitant MDD. About 15% of adult US citizens suffer with irritable bowel syndrome, which also contributes to up to 25% outpatient workload of a gastroenterologist [22, 23]. Thus, it is important to prevent IBS among MDD patients at the onset. On the other hand, given the co-exist of the two diseases, in clinical practice antidepressants including both tricyclic

antidepressants (TCAs) and selective serotonin reuptake inhibitors (SSRIs) are usually used to treat irritable bowel syndrome, the possible pathophysiological mechanisms are alterations in serotonergic signaling or metabolism and disturbance in neurotransmitter-related control of communication between the enteric nervous system and the brain, or called "the brain-gut axis" [22–24]. A potential mechanism that could explain the increased risk of IBS associated with MDD involves the activation of the cingulate region of the limbic system in the cerebral cortex. Drossman et al. utilized functional magnetic resonance imaging and observed that depression can elicit activity in this brain region [25]; Importantly, the limbic system's emotional activity center shares the same anatomical site as the vegetative and endocrine regulatory centers, which govern digestive tract movement and secretion. Consequently, alterations in the function of these centers may occur, leading to changes in neurotransmitters and adreno-corticotropin-releasing factors. These alterations can subsequently impact visceral sensation, intestinal movement, and endocrine function. Thus, this cascade of events may contribute to the onset or exacerbation of IBS symptoms [26].

There are several limitations to our study that should be noted. Firstly, in order to minimize bias related to population stratification, our study was restricted to individuals of European ancestry, limiting the generalizability of our findings to other ancestral backgrounds. Secondly, the identification of IBS cases relied on self-reported data, which may be prone to recall bias and response bias. Additionally, the absence of individual-level data prevented us from conducting stratified analyses by potential effect modifiers such as sex, smoking, and sleep quality. Finally, although we have established a causal link between MDD and IBS, further research is needed to explore the underlying biological mechanisms and potential treatment options.

## 5. Conclusion

To summarize, we found that MDD significantly contributes to the emergence of IBS using Mendelian randomization study. The research findings have potential clinical significance for the care and management of IBS patients. Additionally, efforts should be made to advance the pathophysiology and therapy studies for the two disorders.

## Supporting information

**S1 Table. Details of the GWASs included in the Mendelian randomization.**
(DOCX)

**S2 Table. Single SNPs used in the two- sample Mendelian randomization analysis of the causal effect of major depressive disorder and irritable bowel syndrome.**
(DOCX)

**S3 Table. Leave-one-out sensitivity analysis of causal effects of major depressive disorder and irritable bowel syndrome.**
(DOCX)

**S1 Fig. Funnel plot of single SNPs used in the Mendelian randomization analysis of the effects of major depressive disorder and irritable bowel syndrome.**
(DOCX)

**S2 Fig. Leave-one-out sensitivity analysis for SNPs of major depressive disorder and irritable bowel syndrome.**
(DOCX)

## Author Contributions

**Conceptualization:** Guowei Sun.

**Data curation:** Guowei Sun, Yueyi Jiang.

**Formal analysis:** Guowei Sun.

**Methodology:** Guowei Sun.

**Software:** Guowei Sun.

**Supervision:** Yueyi Jiang.

**Validation:** Guowei Sun.

**Writing – original draft:** Guowei Sun.

**Writing – review & editing:** Yueyi Jiang.

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
