## [Decision Letter · Decision Letter 0]

13 Dec 2023

PONE-D-23-22484Major depressive disorder and irritable bowel syndrome risk: A Mendelian randomization studyPLOS ONE

Dear Dr. Jiang,

Thank you for submitting your manuscript to PLOS ONE. After careful consideration, we feel that it has merit but does not fully meet PLOS ONE’s publication criteria as it currently stands. Therefore, we invite you to submit a revised version of the manuscript that addresses the points raised during the review process.

The reviewers' comments are provided at the bottom of this email. 

We look forward to receiving your revised manuscript.

Kind regards,

Bo Hu, PhD

Academic Editor

PLOS ONE

Journal Requirements:

Additional Editor Comments (if provided):

Please describe the validity of the key assumptions for the MR analysis, especially the effect of potential confounders.

While the results are statistically significant, the effect size found is rather small (i.e., OR=1.01).

What are the columns of id.exposure and id.outcome in Tables S2 and S3? Please annotate them appropriately. Also, the column of sample size is not needed, which can be briefly described as a footnote.

Minor comments: (1) the MAF threshold for SNP selection should be “>=0.3” but not “=0.3”; (2) please clarify whether R2 of 0.1 or 0.001 was used for LD threshold; (2) please explain the dots in Figure 2.

Reviewers' comments:

Reviewer's Responses to Questions

**Comments to the Author**

1. Is the manuscript technically sound, and do the data support the conclusions?

Reviewer #1: Yes

Reviewer #2: Yes

Reviewer #3: Yes

2. Has the statistical analysis been performed appropriately and rigorously? 

Reviewer #1: Yes

Reviewer #2: Yes

Reviewer #3: No

3. Have the authors made all data underlying the findings in their manuscript fully available?

Reviewer #1: Yes

Reviewer #2: Yes

Reviewer #3: Yes

4. Is the manuscript presented in an intelligible fashion and written in standard English?

Reviewer #1: Yes

Reviewer #2: Yes

Reviewer #3: No

5. Review Comments to the Author

Reviewer #1: The authors have explored the causal association of major depressive disorder with irritable bowel syndrome through a bidirectional mendelian randomization study. The design of this study is reasonable, and the manuscript language needs to be further adjusted before publication.

Reviewer #2: (Exclusion hypothesis) (Figs.1). Figs 1 to be written as Fig 1.

Table 1, P val or P or italicized p to be to be standardized throughout the manuscript. 95% to be written as 95% CI.

There were a number of typographical errors in the manuscript.

( Supplementary Tables S1 to be presented as (Supplementary Table S1). For year[1], the gap of the cited reference to be spaced out. This apply to others.

Labelled to be written in the first column for Table S2 and Table S3.

The sentence ‘… individual SNP IVs: MR-Egger regression….’ requires revision.

The definition of IVW could be improved e.g. The inverse variance weighted (IVW) method estimates a causal effect by calculating the slope of a regression line over the associations between the weighted SNP-mean exposure and SNP-mean outcome (oriented to be positive).

For Figure 2, the figure in X and Y axis to be labelled.

p < 5.0 x 10-8 incorrectly labelled.

The format presentation for (p < 5.0 x 10^-8) to follow the format that were presented in Supplementary Tables e.g. p <5.0E-08.

References did not conform to the journal format.

Reviewer #3: Title: Major depressive disorder and irritable bowel syndrome risk: A Mendelian randomization study

This paper utilizes Mendelian randomization to explore the potential causal relationship between major depressive disorder (MDD) and irritable bowel syndrome (IBS). Leveraging data from the Psychiatric Genomics Consortium and the Medical Research Council Integrative Epidemiology Unit, the study employs various Mendelian randomization methods, revealing a significant genetic link between MDD and an elevated risk of IBS. Rigorous analyses, including assessments for heterogeneity and bidirectionality, support the study's findings. Notably, the absence of a genetic link between inflammatory bowel disease and MDD is observed. In conclusion, the study underscores a direct causal association between MDD and IBS, offering valuable insights for clinical management. By incorporating the following change, the paper can potentially be further improved.

Abstract

1. In the Result section of Abstract author using first time IVW in the paper consider defining its full form here

IVW method � Inverse Variance Weighting (IVW) method

Introduction

1. "but there is no known structural abnormalities" might read better as "but there are no known structural abnormalities."

2. "According to a systematic study released in2013" should have a space between "in" and "2013" for proper formatting.

3. "This patients had greater rates of IBS symptoms than those in the control group" needs to be corrected to "These patients had higher rates of IBS symptoms than those in the control group."

4. “ According to a systematic study released in2013, the cost of irritable bowel syndrome treatment ranges from $1,562 to $7,547 year for direct costs and from $791 to $7,737 annually for indirect costs in the United States[10].” This sentence is not clear. Is treatment ranges from $1,562 to $7,547 per year?

Methods

The methods section provides a comprehensive approach to Mendelian randomization (MR) but could benefit from a clearer organization and improved clarity in several areas:

1. The three main hypotheses for an MR study could be presented more clearly, perhaps with a brief explanation for each, ensuring that the reader grasps their significance to the study.

2. The section on data resources is concise but consider providing a brief rationale for choosing the Psychiatric Genomics Consortium (PGC) and Medical Research Council Integrative Epidemiology Unit (MRC-IEU) datasets, highlighting their relevance to the research.

3. The criteria for SNP selection are well-defined, but it might be beneficial to provide a concise justification for these criteria and their significance in the context of the study.

4. The mention of the TwoSampleMR and MRPRESSO packages lacks supportive information or citation.

5. The phrase "version0.5.6" should have a space between "version" and the version number.

Results

1. The author states that in this study, genome-wide significant and independent SNPs were chosen as instrumental variables (IVs) for Mendelian Randomization (MR) analysis. However, the text lacks a clear explanation of how these criteria enhance the reliability of the instrumental variables. Providing a concise clarification on how the stringent criteria for significance and independence contribute to the robustness and validity of the MR analysis would strengthen the clarity and transparency of the methodology.

2. Sensitivity Analysis: Elaborate briefly on why these specific sensitivity analyses were chosen and how they contribute to the robustness of the main analysis.

3. Leave-One-Out Sensitivity Analysis: Consider incorporating a brief interpretation of the results of this analysis.

4. The inclusion of a leave-one-out sensitivity analysis is commendable, providing insight into the stability of the causal effect estimation. However, to enhance clarity, it would be beneficial to briefly define or explain what a forest map is for readers unfamiliar with the term. Additionally, consider specifying the key findings or patterns observed in the forest map that support the conclusion of a reliable and stable causal effect of MDD on IBS, even when individual SNPs are excluded. This would provide a more comprehensive and accessible interpretation of the results for a broader audience.

5. Quality control: Providing a concise explanation of each method (Horizontal polymorphism, heterogeneity analyses, and sensitivity analyses ) and their relevance to quality control would enhance transparency and the overall robustness of the study.

Discussion

The discussion provides valuable insights into the association between Major Depressive Disorder (MDD) and Irritable Bowel Syndrome (IBS) using Mendelian randomization (MR) analysis and large-scale GWAS data. It highlights the significance of the gut-brain axis in connecting gut and brain processes and suggests a causal link between MDD and IBS.

1. However, the discussion could be strengthened by incorporating references to support the mentioned hypotheses and findings. It's essential to provide citations for past studies or theories that have contributed to the understanding of the gut-brain axis and the relationship between MDD and IBS. Referencing existing literature would add credibility to the claims made in the discussion.

2. Finally, the limitations are appropriately acknowledged, but it would be beneficial to elaborate on potential biases or confounding factors that might affect the generalizability of the results. This would add nuance to the interpretation of the study's limitations.

3. The phrase "It is important to prevent IBS among MDD patients at the oneset" contains a typographical error. It should be "onset" instead of "oneset."

Conclusion

1. The discussion effectively emphasizes the clinical implications of understanding the causal relationship between MDD and IBS, advocating for prevention and control strategies. However, explicitly stating the potential clinical impact of the study's findings on patient care and management would provide a more concrete conclusion.

6. PLOS authors have the option to publish the peer review history of their article (what does this mean?). If published, this will include your full peer review and any attached files.

Reviewer #1: **Yes: **Shi-Yang Guan

Reviewer #2: No

Reviewer #3: No

---

## [Author Response · Author response to Decision Letter 0]

2 Feb 2024

Please refer to the document Response to Reviewers.

---

## [Decision Letter · Decision Letter 1]

26 Feb 2024

Major depressive disorder and irritable bowel syndrome risk: A Mendelian randomization study

PONE-D-23-22484R1

Dear Dr. Jiang,

We’re pleased to inform you that your manuscript has been judged scientifically suitable for publication and will be formally accepted for publication once it meets all outstanding technical requirements.

Kind regards,

Bo Hu, PhD

Academic Editor

PLOS ONE

Additional Editor Comments (optional):

Thank the authors for addressing the comments. The quality of the paper was greatly improved. There is only one minor comment regarding space format, which can be corrected without further review.

Reviewers' comments:

Reviewer's Responses to Questions

**Comments to the Author**

1. If the authors have adequately addressed your comments raised in a previous round of review and you feel that this manuscript is now acceptable for publication, you may indicate that here to bypass the “Comments to the Author” section, enter your conflict of interest statement in the “Confidential to Editor” section, and submit your "Accept" recommendation.

Reviewer #2: All comments have been addressed

Reviewer #3: All comments have been addressed

2. Is the manuscript technically sound, and do the data support the conclusions?

Reviewer #2: (No Response)

Reviewer #3: Yes

3. Has the statistical analysis been performed appropriately and rigorously? 

Reviewer #2: (No Response)

Reviewer #3: Yes

4. Have the authors made all data underlying the findings in their manuscript fully available?

Reviewer #2: (No Response)

Reviewer #3: Yes

5. Is the manuscript presented in an intelligible fashion and written in standard English?

Reviewer #2: (No Response)

Reviewer #3: Yes

6. Review Comments to the Author

Reviewer #2: The authors have put in significant efforts to address the comments.

Lines 232-234, Lines 289- 297, the rows spacing is to be consistent with others.

Reviewer #3: Thank you for addressing all the concerns raised during the revision of your paper. Your efforts have significantly improved its quality and clarity.

7. PLOS authors have the option to publish the peer review history of their article (what does this mean?). If published, this will include your full peer review and any attached files.

Reviewer #2: No

Reviewer #3: **Yes: **Sarita Poonia

---

## [Editor Report · Acceptance letter]

6 Mar 2024

PONE-D-23-22484R1 

PLOS ONE

Dear Dr. Jiang, 

I'm pleased to inform you that your manuscript has been deemed suitable for publication in PLOS ONE. Congratulations! Your manuscript is now being handed over to our production team.

Kind regards, 

on behalf of

Dr. Bo Hu 

Academic Editor

PLOS ONE